# Simulating Urban Element Design with Pedestrian Attention: Visual Saliency as Aid for More Visible Wayfinding Design

**Gwangbin Kim** [1] , **Dohyeon Yeo** [1], **Jieun Lee** [1] **and SeungJun Kim** [1,2,*]

1    School of Integrated Technology, Gwangju Institute of Science and Technology, Gwangju 61005, Republic of Korea
2    AI Graduate School, Gwangju Institute of Science and Technology, Gwangju 61005, Republic of Korea
*    Correspondence: seungjun@gist.ac.kr; Tel.: +82-62-715-5352

**Abstract:** Signs, landmarks, and other urban elements should attract attention to or harmonize with the environment for successful landscape design. These elements also provide information during navigation—particularly for people with cognitive difficulties or those unfamiliar with the geographical area. Nevertheless, some urban components are less eye-catching than intended because they are created and positioned irrespective of their surroundings. While quantitative measures such as eye tracking have been introduced, they help the initial or final stage of the urban design process and they involve expensive experiments. We introduce machine-learning-predicted visual saliency as iterative feedback for pedestrian attention during urban element design. Our user study focused on wayfinding signs as part of urban design and revealed that providing saliency prediction promoted a more efficient and helpful design experience without compromising usability. The saliency-guided design practice also contributed to producing more eye-catching and aesthetically pleasing urban elements. The study demonstrated that visual saliency can lead to an improved urban design experience and outcome, resulting in more accessible cities for citizens, visitors, and people with cognitive impairments.

**Keywords:** design feedback; visual saliency; wayfinding design; urban planning





## 1. Introduction

Urban-form elements and architectures are designed to capture attention or to be easily neglected. For example, landmarks such as buildings or monuments affect the impression of a city or assist wayfinding and thus should be conspicuous [1]. Similarly, navigational signs should be easily noticeable to provide route information at key moments or locations. In contrast, urban elements that play a secondary role in the landscape should blend in visually with their surroundings. For instance, artificial structures such as wind power generators may negatively impact the urban landscape and should therefore be inconspicuous [2]. Thus, attention drawn or distracted by urban elements and their impact on the overall landscape should be carefully considered during the urban design process, including the analysis, creation, placement, and evaluation of urban elements.

Attention-aware design is particularly important in wayfinding design—a subset of urban design that involves organizing navigational and environmental information to construct signage and other cues that inform users of where they are, where they should go, and what they can or cannot do. To achieve this, signage must be sufficiently visible [3]. When signs have poor information content or a poor presentation method, they can be missed, resulting in wayfinding failure [4]. Additionally, a signage system is not an independent visual entity but a complex component of urban planning, as wayfinding design exists at the nexus of graphic design, urban and architectural design, and landscape planning [5]. Thus, the value of signage should be carefully assessed individually and in the context of placement and information delivery, similar to other urban elements.

Well-designed wayfinding is essential for people's daily lives, as it helps people move around cities to work, play, or engage in other activities. Urban accessibility is a vital human right [6], and people's mobility and visits to urban spaces can affect their health [7]. However, identifying one's position in a physical place and navigating to another destination may be difficult for those in unfamiliar places who must rely on external information from signs and landmarks [4]. Likewise, wayfinding may be hindered for older adults with diminished physical and cognitive capabilities [8]. Effective wayfinding design with clear signs and landmarks can facilitate navigation for all people, including visitors and the elderly, making cities more accessible and inclusive.

However, designers face the challenge of predicting which elements or objects will be conspicuous [9]. This is particularly difficult for wayfinding design that involves both graphic and urban design characteristics. Signs should stand out, as they may be the most informative urban elements. However, these aspects are sometimes at odds: while graphic artifacts such as posters may be evaluated according to the layout, wayfinding signs should be evaluated according to their compatibility with their surroundings. Additionally, wayfinding signs and other urban elements are often semi-permanent or difficult to modify once constructed. It is thus important to simulate and design them in a way that attracts attention from the start through iterative design processes.

To address this issue, researchers have involved citizens in the design process, allowing them to have a say in the landscapes they will ultimately use. This engagement is usually focused on the initial or final stages of design to assess whether a design is effective. A similar approach has been applied to wayfinding design, with methods such as surveys, workshops, and interviews for gaining feedback on proposed designs [10–12]. Newer methods, such as crowd rating [13] and eye tracking [14,15], have also been used for quantitative evaluation. However, these methods are limited, as designers cannot see the impact of design changes in real time. Visual saliency can provide a solution, as it can give immediate feedback on how users engage with the design, allowing for real-time simulation during the walking and navigation of pedestrians.

In this work, we propose a new approach to urban and wayfinding design that incorporates visual-saliency prediction. The method allows designers to evaluate and iterate their designs with a machine-learning algorithm that predicts how pedestrians will engage with their designs (Figure 1). We performed a sanity check on our tool to assess its usability and gauge the experience of the designers using it. The effectiveness of the method was then determined by assessing the overall improvement in design quality. The ability to anticipate people's visual attention in different contexts can help create more navigable cities through the design of accessible and conspicuous wayfinding signs.

### 1.1. Related Works

1.1.1. Urban Design Tools

Urban design has advanced from paper-based maps to computer simulation tools such as geographic information systems (GISs), which allow a multifaceted analysis of planning projects using geo-referenced data [16]. These tools, such as CityEngine, allow accurate planning through procedural modeling of urban elements [17], but they are complex and require extensive training to use [18]. Furthermore, they have a limited ability to simulate interactions between elements in spaces [18,19]. Game engines such as Unity 3D are increasingly being used as urban design tools because of their accessibility, ease of use, and convenient design experience. They also minimize the limitations of computer-aided design (CAD) tools that require expensive software licenses or technical and geographic expertise [20]. For instance, Unity 3D offers rapid importing of ready-made models, materials, and textures with photorealistic rendering. The tool is also highly compatible with existing CAD/GIS software data [21] and allows interactive simulation of navigation and interaction with objects [22]. Thus, we selected this game engine as a basic tool for wayfinding design to support realistic simulation of pedestrian interactions.

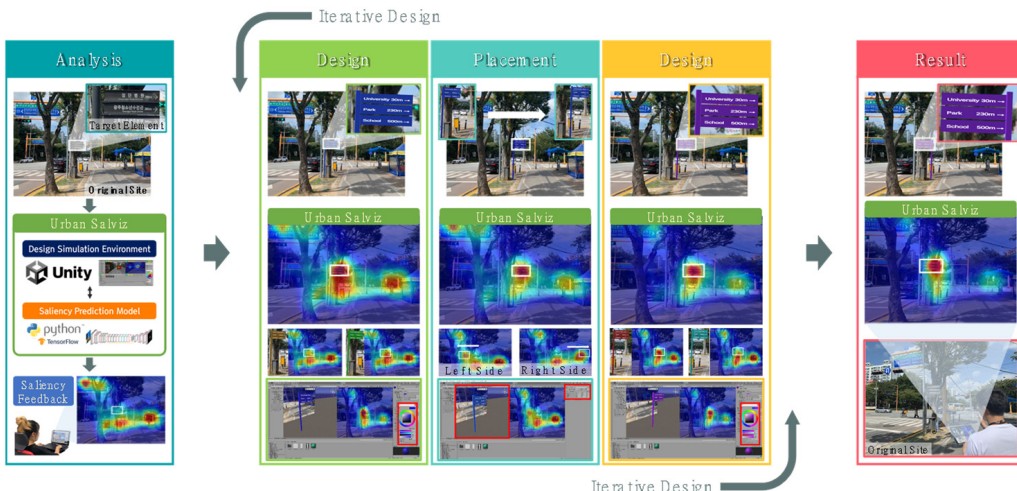

**Figure 1.** Incorporating saliency to iteratively enhance the conspicuity of a sign. Pedestrian attention is visualized throughout the design process, including site analysis, implementation, iteration, and installation. In this case, desginers can simulate the amount of attention paid to wayfinding signs with different color and position (boxed elements in the mixed reality environment).

### 1.1.2. Tools and Methods to Promote Iterative, Attention-Aware Design

Design and creativity support tools advance innovation and discovery by supporting the design process [23]. However, most professional design tools focus on either the very early or very late design phases, such as pre-ideation, background research, idea generation, or final evaluation, and focus less on other stages of design in which iterative crafting and modification occur. Indeed, only 6% of the surveyed methods supported iterative processes [24]. Despite its importance in yielding high design quality, in cases of urban design, the actual crafting process (outside of the context of listening to citizens' initial or final decisions) has rarely been aided by design support tools.

While catching people's attention is essential in urban, landscape, and wayfinding design [2,11,25,26], one of the most difficult problems for designers is predicting where people will focus their attention [9]. While designers previously relied on qualitative methods such as a workshop or an interview, they are increasingly including quantitative measures in the design evaluation feedback. For example, eye tracking provides quantitative attention data but requires dedicated hardware and a sufficient number of human subjects for obtaining valid and reliable results. In recent studies, crowd feedback has been employed for design reflection [27–30], and eye gaze can be crowdsourced via the web [31]. However, these methods require numerous participants, and the feedback response time for a single iteration is at least a few minutes. In addition, because the current methods for gaze crowdsourcing are designed for static images, designers cannot simulate the effects of possible interactions when scenes change while pedestrians walk, wander, and turn. To solve this problem, we replace the eye tracker with a saliency map generated by machine-learning models, which simulate human attention and can be used to iterate and improve the design more efficiently. The proposed method and previous methods for gathering design evaluation and feedback for urban design are presented in Table 1.

### 1.1.3. Visual Saliency as Aggregated Attention to Support Urban Design

Human visual and cognitive systems allocate more attention to things that stand out from the rest of the view, and this effect is known as visual saliency. Visual-saliency models simulate human visual processing to predict the amount of attention paid to each area in an image, which is known as fixation prediction [32]. These models can be classified as stimulus-driven or data-driven. Stimulus-driven models are based on low-level features, such as color, size, and shape [32,33]. In contrast, data-driven models are trained in an end-to-end manner to evaluate saliency in an image as a regression problem [33],

and they have outperformed stimulus-driven models with advancements in deep neural networks.

Owing to their remarkable performance, visual-saliency models have been applied to various design tasks, e.g., posters [34,35], web interfaces [9,35,36], and mobile interfaces [37,38]. However, the effectiveness of these tools for enhancing design quality remains unclear. This is mainly because the previous research had a different goal [34] or was not based on user studies [36–39], the effect of saliency prediction was diluted by involving additional assistant utilities [9], or the result of a single-item rating was reported without statistical evidence [25]. Additionally, few researchers have applied visual-saliency models to support the design of urban landscape elements and wayfinding. A landscape observation experiment performed by Dupont et al. [25] indicated that saliency maps are strongly correlated with human focus maps, suggesting they can be used to predict human observations in urban landscapes. In the remainder of the paper, our application of visual saliency as urban design feedback is explained. We investigated its effect on the quality of the resulting design.

**Table 1.** Advantages and disadvantages of the proposed method and previous methods for urban design feedback.

| Relevant Works | Feedback Methods | Design Target | Advantages | Disadvantages |
|---|---|---|---|---|
| Luo et al. (2022) [40] | Workshop | Urban park | Promote interactive collaboration among participants | Mainly applicable to the initial and final design stages |
| Marthya et al. (2021) [41] | Interview | Transit corridor | Can gather the various stakeholders' opinions Contains the detailed requirements of the citizens | Collecting the full response is time-consuming Results are not quantifiable |
| Seetharaman et al. (2020) [11] | Photovoice | Landmark | Can capture instant images of the urban design | Spatial contexts of the design are not considered in detail |
| Saha et al. (2019) [13] | Crowd rating | Sidewalk | No need for face-to-face experiments Fast response from online workers | Cannot guarantee the feedback data quality Crowdworkers are not always the dwellers |
| Vainio et al. (2019) [14] | Eye tracker | Urban scenery | Can collect quantitative human attention including unconscious perception | Requires a dedicated hardware device Difficult to interpret the large amount of data |
| This work (2023) | Visual saliency | Wayfinding sign (user study) Building, poster (tool compatibility) | Real time, no need for human participants Can simulate 3D dynamic interactions | Difficult to observe individual preferences Cannot gather detailed comments |

### 1.2. Contribution Statement

Our study contributes to the fields of user interface (UI)/user experience (UX) design, urban and landscape planning, and human factors and ergonomics as follows:

- We introduced the concept of saliency-guided iterative design practice to the field of urban design, specifically in the domain of simulating wayfinding design.
- We demonstrated that providing visual-saliency prediction as design feedback can improve the design.
- We showed that saliency feedback can be particularly effective for designing accessible wayfinding for older adults with cognitive decline.

- We provided a detailed system implementation and reusable code for a three-dimensional (3D) urban design simulation tool with visual saliency and pedestrian spatial interaction.

## 2. Methods

### 2.1. Study Overview

The objective of this study was to examine the impact of visual-saliency prediction on the quality of the outcome in an urban design process. Specifically, we aimed at answering the following research questions:

- RQ1: Does saliency prediction feedback help designers create better urban designs?
- RQ2: In which environments are the designs produced with saliency prediction particularly effective?
- RQ3: Which end users will the saliency-guided design method benefit the most?

Through literature reviews, we established research hypotheses for the aforementioned questions.

#### 2.1.1. RH1: Saliency Prediction Helps Designers Create Better Designs by Helping Them Anticipate the Elements to Which Pedestrians Will Pay Attention

How a design attracts people's attention is an important factor in urban environments [14]; however, most designers struggle to simulate this in advance. A graphic design tool by Lee et al. [9] that includes a saliency map was found to improve web design. Lee et al. [9] tested their design tool as a whole, including various utilities for automatic evaluation and design recommendation. Cheng et al. [35] reported that providing an attention map to users improves the design, although their study involved 10 participants, and a statistical analysis was not conducted. Because urban design involves environmental and graphical contexts, we hypothesized that providing visual-saliency prediction during the design of urban elements would enhance the quality of the designs.

#### 2.1.2. RH2: A Design Produced with Saliency Prediction Is More Effective for Capturing Attention in Urban Areas than in Rural Areas

The visual and structural characteristics of environments affect the ease of wayfinding [42,43]. Urban environments contain more visually distracting elements, such as buildings, signs, and other spatial elements, than the rural countryside. For example, Costa et al. [44] found that drivers fixate on road signs significantly more in rural areas than in urban areas. The increased complexity and likelihood of distraction pose another challenge to predicting where users will pay attention. In addition, with regard to visual complexity and distraction, the design outcome should compete for end users' attention—particularly in urban areas. Thus, we hypothesized that end users would find the designs produced using our method more effective for urban areas than for rural areas.

#### 2.1.3. RH3: Elderly People with Subjective Cognitive Decline (SCD) Prefer Designs Created with Saliency Feedback More Strongly Than Young Adults and Elderly People without SCD

Age is the most significant risk factor for decline in cognitive functions, such as attention, working memory, processing speed, and verbal and visual explicit memory [45]. Two strong indicators of age-associated cognitive decline are SCD, which refers to self-reported decline [46], and difficulty with spatial navigation [45,47,48]. Cerman et al. reported that 68% of subjects with SCD complained about their spatial navigation abilities [49]. Thus, for older adults, visual stimuli that do not capture attention properly may not be fed to working memory from sensory memory. Thus, salient design promotes faster recognition of road signs and landmarks, supporting spatial navigation. Therefore, we hypothesized that older adults with SCD would be particularly likely to prefer designs produced with saliency feedback compared with other demographic populations.

To verify the hypotheses, we created Urban Salviz—a mixed-reality urban design simulation tool—using the Unity 3D game engine and divided 32 designers into

two groups: one with access to the saliency prediction function and one without. In addition, we performed a sanity check to assess the effects of saliency prediction on the usability and design experience of the tool. Finally, the design outcomes produced by both groups were evaluated by 95 people to assess the impact of saliency-based feedback on the quality of the design outcome (Figure 2).

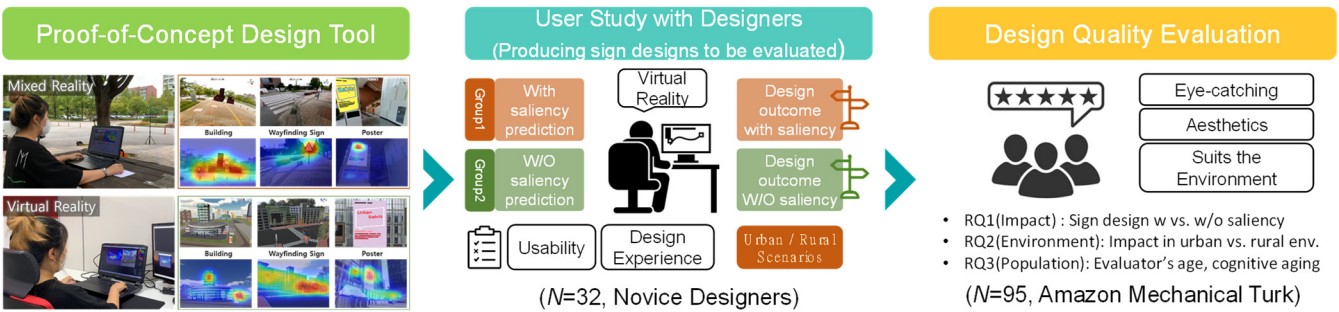

**Figure 2.** Overview of the study design. We built a proof-of-concept urban design tool to assess the effect of providing saliency feedback during the urban design process. For a sanity check, we recruited 32 designers who used the tool for design to test its usability and design experience both with and without saliency prediction. The produced design outcomes were evaluated by 95 participants to assess the impact of saliency prediction on the design quality (RQ1) according to the environment (RQ2) and the evaluator's age (RQ3).

### 2.2. Urban Salviz

#### 2.2.1. Tool Overview

We built a proof-of-concept urban design tool allowing users to model 3D objects, position them in graphical environments, and edit their design elements, such as text contents, figures, colors, size, and rotation (Figure 3). We also included 3D interactions in spaces (e.g., walking, jumping, and head rotation) in the design tool. Additionally, the tool includes example sign designs for enhancing the design quality and promoting designer creativity [50].

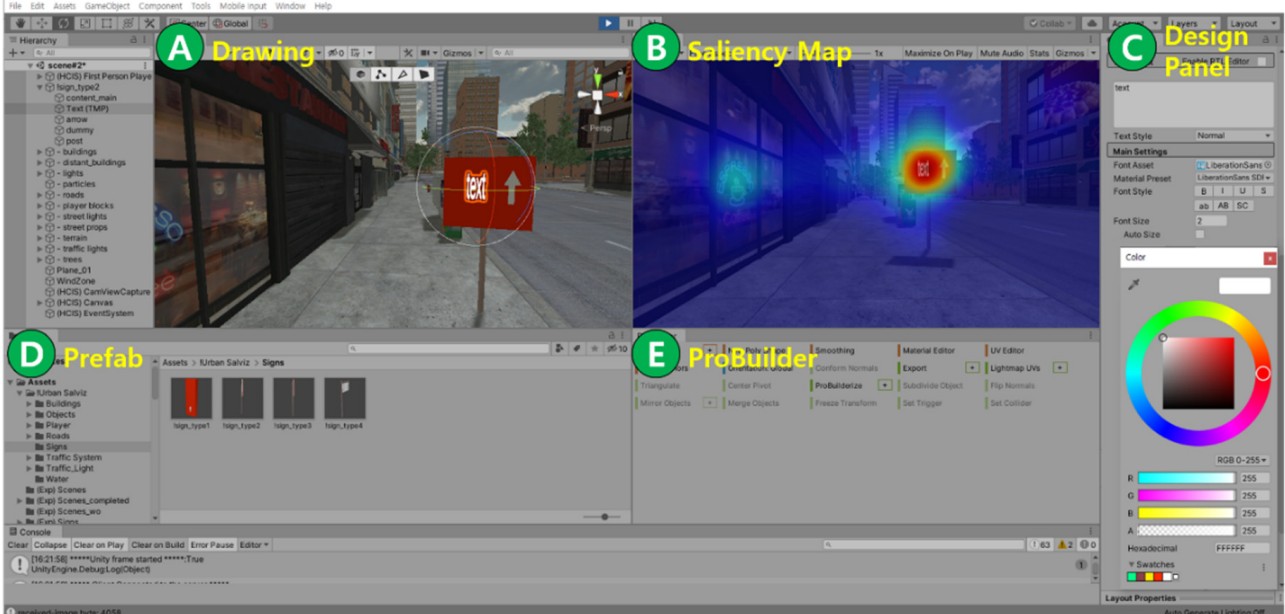

**Figure 3.** Interface outline of our urban design tool, which is based on Unity 3D and allows users to create 3D urban elements either from a half-made model (**D**) or from scratch (**E**) and fine-tune the design elements (**C**). Users can simulate the location, orientation, and size of the designed object (**A**) while checking a virtual pedestrian's viewpoint or visual attention (**B**).

A.  *Drawing.* Users can alter the design of the urban elements in the *Drawing* Table. This tab is based on the Scene view of Unity 3D; thus, it includes the transform components of the tool. Users can transform objects in the X, Y, and Z directions (Figure 4, Move); rotate objects around the X, Y, and Z axes (Figure 4, Rotate); scale objects along the X, Y, and Z axes (Figure 4, Scale); and stretch objects along the X and Y axes (Figure 4, Rect). Users can either select virtual-reality (VR) environments or load a camera input for video through mixed-reality applications.

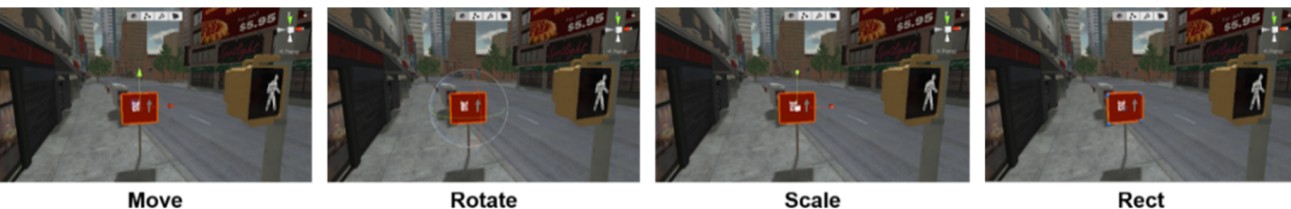

**Figure 4.** Object transform functions. Users can move, rotate, scale, and rect (two-dimensionally scale) target objects.

B.  *Saliency Map.* The *Saliency Map* panel (Figure 3B) offers real-time visual-saliency predictions from a virtual pedestrian viewpoint. The visual-saliency prediction mimics fixation prediction and helps designers to determine where users will pay the most attention to. The board reflects every change in the *Drawing* tab (Figure 3A) and *Design Panel* (Figure 3C), updates the visual saliency in real time, and can be turned on or off with the space bar. Users move the virtual pedestrian's pose and perspective with a mouse or keyboard. In our user study, this function was not used by the control group, whereas the experimental group could freely turn the saliency map on or off.

The implementation of the function involves capturing a real-time video of the graphical scene view (*Drawing* tab, Figure 3A) through the TCP protocol and feeding it into the Python environment. The Python part then employs a deep-learning model to generate a saliency map of the scene view, which is imported into the *Saliency Map* (Figure 3B) in the Unity 3D environment.

To accurately predict the saliency of the cityscape, we incorporated MSI-Net (Figure 5) [19] owing to its performance on the MIT Saliency Benchmark [51,52]. MSI-Net uses an encoder–decoder structure with convolutional layers to capture features at multiple spatial scales. The encoder network processes the input image and extracts relevant features, and the decoder network uses the encoded features to generate the final predicted saliency map. The model used in this study was able to assess the visual saliency of arbitrary urban and rural environments with wayfinding signs by learning the aggregated visual attention paid to 10,000 images of both types of environments from the SALICON dataset [53] (Figure 6).

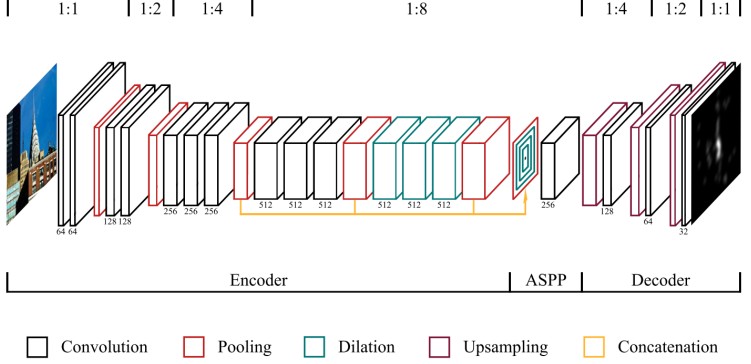

**Figure 5.** Architecture of MSI-Net [19]. The encoder module extracts the multiscale features with multiple convolutional layers, and the decoder module produces the predicted saliency map.

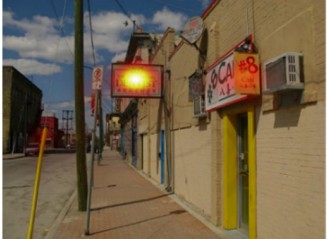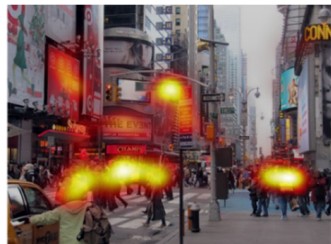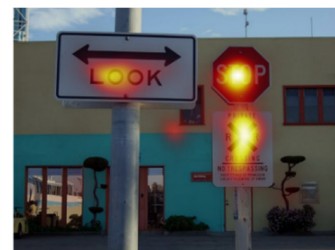

**Figure 6.** Image examples from the SALICON dataset [53], which includes urban and rural landscape images that contain wayfinding signs.

The grayscale attention from the model was transformed into a color heatmap. We used the heatmap as a visualization tool for user attention, as it has been the most popular since their introduction to gaze visualization [54,55]. Compared with scan paths, which are not suited for the visualization of multiple users [55], a heatmap presents aggregated gaze data more clearly by combining the gaze points of numerous viewers. We converted the grayscale heatmap value rage of 0–255 into a jet colormap that ranged from blue to red—passing through green, yellow, and orange—to visualize the saliency map for users (Figure 3B).

C.   *Design Panel*. The *Design Panel* is used to design and adjust the visual details of the crafted elements. It is based on the Unity 3D inspector window, which allows users to view and edit the properties and settings of game objects, assets, and prefabs. When a user clicks an object to edit in the *Drawing* tab, the content of the design panel changes to display the object's visual attributes object. Users can add text, change the font size, and adjust the color of each element of the selected object. Users can also move, rotate, and scale objects in a numerical manner.

D.   *Prefab*. A *Prefab* is a saved reusable game object consisting of multiple object components that functions as a template. We developed four prefab models of wayfinding signs with different designs (Figure 3D). Each prefab consisted of a sign panel, a text box, an arrow, and a pillar. Designers can browse example signs and can also start from a half-made design by loading the existing Unity prefab sample models via dragging and dropping.

E.   *ProBuilder*. This tool allows users to create urban elements with custom geometries by constructing, editing, and texturing basic 3D shapes. *ProBuilder*—a hybrid 3D modeling and level design tool optimized for creating simple geometries—is utilized for this. With the ability to select any face or edge for extrusion or insetting, users can create objects with various shapes.

2.2.2. Implementation Notes

Urban Salviz was built and run on a computer with an Intel CPU (Xeon E5-1650 v4@3.60 GHz), 112 GB of RAM, and a GTX 1080Ti GPU. The saliency prediction and heatmap color mapping were computed using an independent Python-based program, and the results were transferred to the 3D design tool through TCP socket communication (Figure 7). Thus, the function can be added to any design tool with a proper communication configuration. The software implementation source code used in the paper is available at the author's repository: https://github.com/GWANGBIN/Urban-Salviz (accessed on 19 December 2022).

*2.3. User Study with Designers (Sanity Check of Usability and UX for Crafting Road Signs)*

To test the viability of the tool, we assessed its usability and design experience with and without saliency prediction.

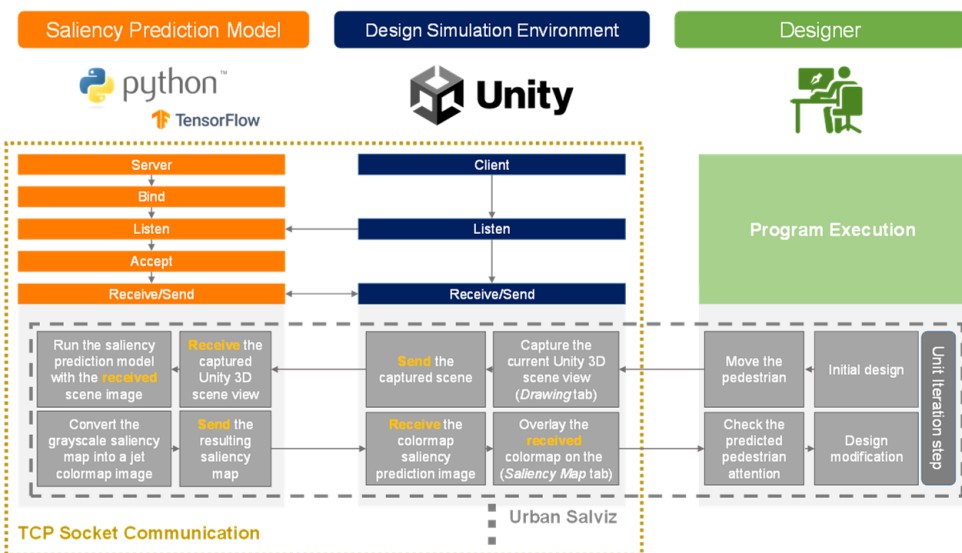

**Figure 7.** Implementation architecture of Urban Salviz. The design simulation part is based on Unity, and the saliency prediction is based on Python TensorFlow. Both parts continuously send and transfer the targets and resulting images to provide seamless iterative feedback.

### 2.3.1. Procedure

Initially, participants completed pre-experiment questionnaires to provide their demographic and background information, including name, age, gender, and design experience. Then, the experimenter demonstrated each function of the tool for 20 min, following fixed guidelines to provide the same information to all the participants. After the description, the participants practiced using the tool until they felt sufficiently confident to design with it (20 min on average). Then, the participants proceeded to the main experiment, which involved creating a road sign with our design tool for six scenarios: three urban and three rural environments (Figure 8). As the design outcome produced in the user study was to be rated in the design quality evaluation study, we set both rural and urban scenarios to see in which environment the design with saliency feedback is particularly effective (RQ2). We shuffled the order of the six scenarios via Latin square counterbalancing. For each scenario, participants were asked to design a sign that is "easy to find (eye-catching)", "aesthetically pleasing", and "suits the surrounding scenery".

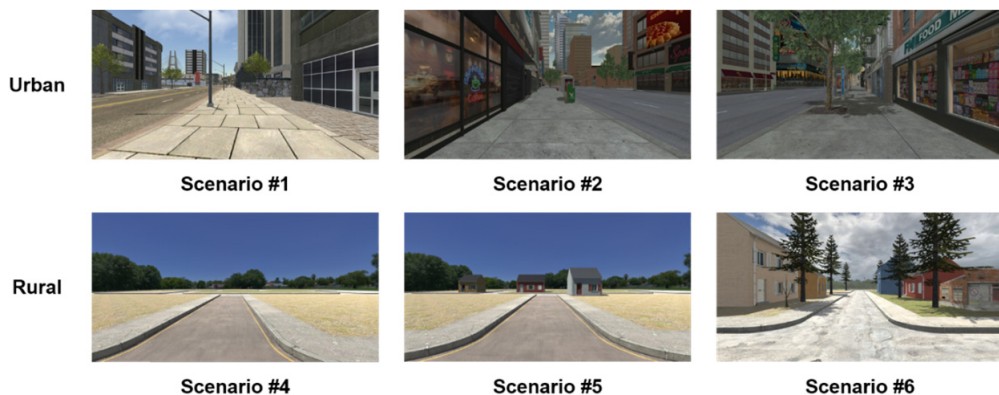

**Figure 8.** Six experimental scenarios (three urban environments and three rural landscapes).

### 2.3.2. Questionnaire Assessing Usability and Perceived Design Experience of Tool

After the design task, participants responded to post-experiment questionnaires involving a system usability scale (SUS) [56]. Because the SUS mainly asked about the overall ease of use of the tool, we included additional items on perceived helpfulness ("I felt that using the tool was helpful in performing the design task"), efficiency ("I felt that the

tool supported my design task in a more efficient way"), and effectiveness ("I view the tool to be effective in producing a wayfinding design of quality") to capture the design experience using a five-point Likert scale (1: "strongly disagree"; 2: "disagree"; 3: "neutral"; 4: "agree"; 5: "strongly agree"). The experiment ended with open-ended questions and post-experiment interviews on the pros and cons of the tool, as well as suggestions for improvement. The average duration of the experiment for each participant was 2 h.

### 2.3.3. Participants

We tested the tool with 32 participants (13 female) aged 19 to 32 years ($M = 23.2$, $SD = 3.5$), via advertisement. Each participant was randomly assigned to either the experimental group or the control group. The experimental group could use all the functions of the tool, and the control group could use all the functions except the saliency prediction to test RQ1 by evaluating the design outcome. Because the two groups performed the same task, except for the saliency prediction, we selected a between-subjects design to avoid the learning effect. In selecting participants, we focused on novice designers because they tend to have difficulty expecting where users will look and anticipating the quality of their design, which is essential for iterative design processes [9,57]. In previous studies, designers with little to no professional experience [9,50,58] or those with 1–5 years of experience were labeled as novices [57]. In our study, to minimize individual differences, participants consisted of designers with less than 3 years of experience (i.e., novice designers).

### 2.4. Design Quality Evaluation (Impact of Saliency Feedback on Design Outcome)

We compared the quality of the sign designs produced by both control and experimental groups in the user study (w vs. w/o saliency feedback) to evaluate the impact of saliency feedback (RQ1), as assessed by crowd workers. Crowdsourcing is widely used for design feedback, including critiques for visual design [59,60], UI design [52], and urban landscapes [61]. Amazon Mechanical Turk (AMT) is widely adopted because it provides accurate results for location-based tasks requiring human intelligence [62]. It has been used to evaluate various urban planning projects (e.g., playability [20,63], livability [64], safety [20,64], and comfort [20] of urban areas) with crowd ratings.

### 2.4.1. Procedure

We recorded 192 design cases from the user study involving designers (32 designers × 6 scenarios) in short video clips to simulate pedestrian walking scenarios so that wayfinding signs could be evaluated in a spatiotemporal context. The videos were recorded from the perspective of a virtual pedestrian walking at 4 km/h past a road sign (Figure 9).

To ensure a fair and consistent comparison, the virtual pedestrian always started at the same distance from the signs. Each video clip lasted approximately 10 s, and the design cases were presented in a random order. In the first 3 s of each video, the following task description was presented, "You're on your way to the [*DESTINATION NAME*]. Find the road sign to the destination in the video and evaluate its design." We then used AMT to recruit participants online [65] for design evaluation. To quantitatively assess the designs, we employed crowd ratings, which are widely used for crowdsourced design evaluation [9,50,66,67].

Because we were concerned about the possibility of malicious or inattentive participants failing to carefully read the questions or responding without watching the full video, we added six "dummy" videos followed by three-question probes as instructional manipulation checks [68,69]. Instructional manipulation checks are effective when they are not distinctive from normal questions at a glance [70,71]. The thumbnail, start, and end of the "dummy" videos were identical to those of the "normal" video clips. However, the instruction in the "dummy" videos was to *"Choose the leftmost answers to the following three questions"* throughout the main portion of the videos (Figure 9). Participants who did not watch the video could not distinguish between the "dummy" and "normal" videos and thus would provide answers other than "1/1/1" to the instructional manipulation check.

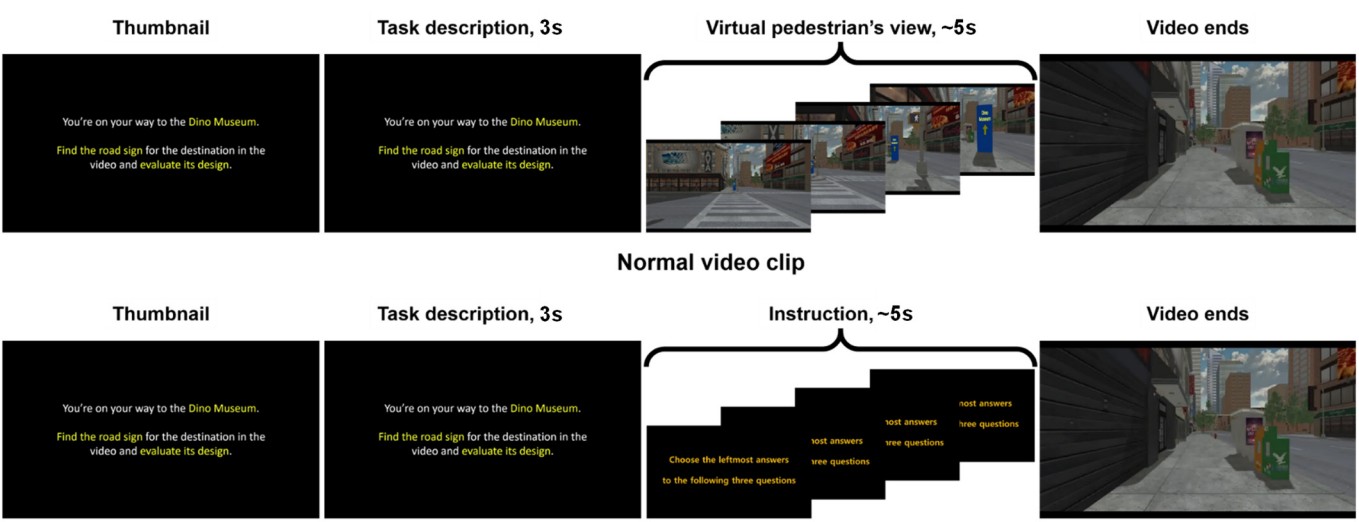

**Figure 9.** Video clips used for crowd design evaluation (upper: normal video clip; lower: instructed question). The main portion of the instructional manipulation check was intended to identify malicious or inattentive participants.

### 2.4.2. Questionnaire for Assessing Quality of Wayfinding Signs

Participants first answered a demographic questionnaire involving age, gender, and design experience. In one questionnaire item, which has been used in previous studies [46,72,73], they indicated whether they were suffering from SCD. A total of 11 participants who were ≥60 years old reported suffering from SCD. Then, the participants started the video-based surveys. After watching each video clip, participants replied to three questionnaire items for assessing the quality of sign design, i.e., "*easy to find (eye-catching)*", "*aesthetic value*", and "*suits the surrounding environment*", on a seven-point Likert scale. All question items were given with detailed rubrics to support the crowd evaluation [74].

To reduce participant fatigue from repetitive tasks, we consolidated several aspects of sign quality into three items (see Table 2). The first item, i.e., *easy to find*, encapsulates key factors such as conspicuity, distinctiveness, intrigue, legibility, identifiability, detectability, and recognizability [75]. It also reflects the Von Restorff effect, which suggests that an object that stands out from its surroundings is more memorable [76]. The second item, i.e., *aesthetic value*, includes aspects such as graphics, simplicity, and completeness. It indicates how aesthetically pleasing a sign is, which affects the overall usability [77,78]. The third item, i.e., *suits the surrounding environment*, considers the isolation effect in the context of similarity [76] to prevent the appraisal of designs that interfered with existing landscapes. These measures were also used by Fontana [79], who summarized the psychophysics of signs as conspicuity, aesthetics, and environmental harmony. Some of the surveyed items were not included because they evaluate the signage system of a country (e.g., traffic or other signs that are under strict regulations) rather than individual designs.

### 2.4.3. Participants

We recruited 227 people with Human Intelligence Task approval rates of >90% [80] from AMT. A total of 127 participants provided incorrect answers in the instructional manipulation check and were thus excluded from the study. Five other participants who passed the instructional manipulation check were rejected for skipping questions. Finally, 95 participants (57 female) aged 23–74 years ($M = 43.6$, $SD = 15.5$) remained eligible for our study. The benefit of using a large participant group in the method allowed us to analyze the results according to different population groups of the evaluators (RQ3). We classified the participants ≥60 years old as the older group ($N = 23$, $M = 65.2$, $SD = 4.3$),

and the others as the younger group ($N = 72$, $M = 36.8$, $SD = 13.2$). The classification was based on the relationship between age and spatial navigation; i.e., the decline in navigation ability becomes distinctive at an age around 60 [81]. Depending on the ages and answers regarding SCD, the participants were divided into two additional groups: the elderly with SCD ($N = 11$) and the others ($N = 84$).

**Table 2.** Questionnaires referenced for assessing the quality of sign designs.

| This Work | Fontana. (2005) [79] | Martin et al. (2015) [82] | Mishler and Neider (2017) [83] | Yang et al. (2019) [84] | Na et al. (2022) [85] |
|---|---|---|---|---|---|
| Easy to find (eye-catching) | Conspicuity | Intrigue | Distinctiveness Isolation | Legibility Identifiability | Detectability Conspicuity Recognizability Legibility Comprehensibility |
| Aesthetic value | Aesthetics | Graphics | Simplicity | Simplicity | |
| Suits the environment | Environmental harmony | Influence | Consistency | Accuracy Non-interference | |
| N/A | | Accessibility | Reassurance | Continuity Completeness | |
| | | Information | | Intelligibility | |

## 3. Results

### 3.1. Sanity Check to Verify Usability and Design Experience of Tool

3.1.1. Saliency Prediction can Be Integrated with the Urban Design Tool without Sacrificing Usability

Both groups (with and without the saliency prediction function) reported that the urban design tool had acceptable usability (SUS score > 70 [86]; Figure 10, left). While Urban Saviz with the saliency function activated was perceived to be more usable ($M = 71.88$, $SD = 13.34$) than the tool without visual-saliency prediction ($M = 71.41$, $SD = 7.47$), the difference was not statistically significant ($U = 101$, $p = 0.30$, $r = 0.18$). This indicates that both with and without the saliency prediction function, the tool can be used as a research platform. The results also imply that the saliency prediction function can be integrated into existing tools without affecting their overall usability.

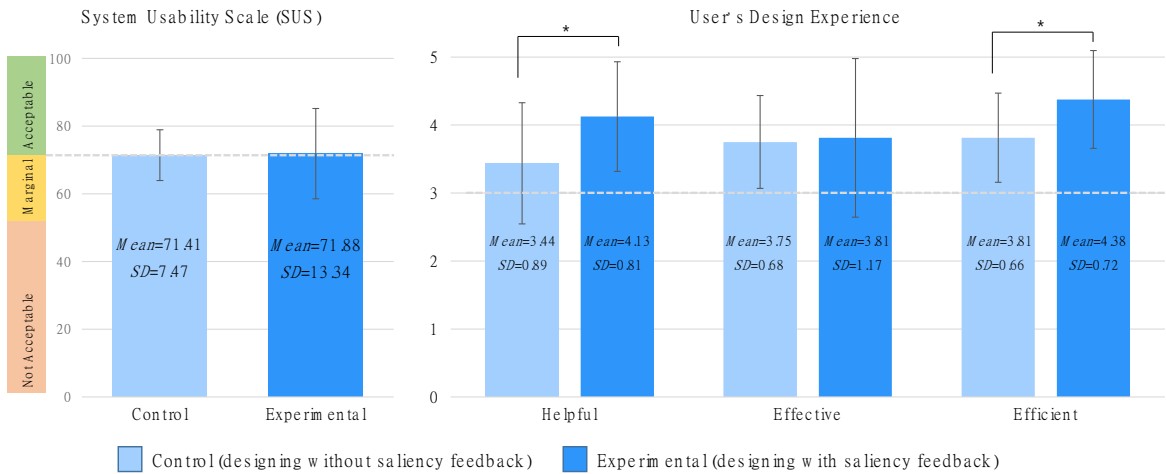

**Figure 10.** SUS scores and users' design experience for the experimental group with saliency prediction and the control group without saliency prediction (* $p < 0.05$). The design process with the saliency map was perceived to be significantly more helpful and efficient than that without the saliency map.

### 3.1.2. Designers Perceived the Tool to Be More Helpful and Efficient When They Were Provided with Saliency Feedback

While there was no significant difference in the overall usability, incorporating saliency prediction functions provided a better design experience (Figure 10, right). The designers who used all the functions, including saliency prediction (*Mean Rank* = 20.03) perceived the tool to be more helpful than those who did not use the saliency function (*Mean Rank* = 12.97), with $U = 71.5$, $p = 0.016$, and $r = 0.43$. Designing signs with saliency feedback (*Mean Rank* = 19.88) was perceived to be significantly more efficient than that without visual saliency (*Mean Rank* = 13.13), with $U = 74$, $p = 0.023$, and $r = 0.40$. However, there was no statistically significant difference for the "effective" item between the experimental group (*Mean Rank* = 17.41) and the control group (*Mean Rank* = 15.59), with $U = 113.5$, $p = 0.55$, and $r = 0.011$. This result partly confirms RH2, i.e., that saliency feedback provides users with a better design experience with regard to perceived helpfulness and efficiency.

In summary, the saliency prediction function provides users with a more helpful and efficient design experience. Participants' responses in the interviews revealed how providing saliency prediction positively affected the design experience. Participants stated that the pedestrian attention prediction function was intuitive (P1, P3, P12), helpful (P1, P3, P4, P5, P12, P14, P19), and efficient (P1, P3, P5, P18) because they could see how their design attracted or distracted user attention. Participant P1 summarized this experience as follows: *"The attention prediction was very helpful as it worked as real-time feedback. I could see which object and how much it attracted pedestrian attention every time I modified design elements (e.g., color, position, sizes, etc.). If it weren't for the attention prediction feedback, I wouldn't have iterated design that much, not knowing how good or bad my design is."*

### 3.2. The Saliency Feedback's Impact on the Quality of the Design Outcome

Each person's evaluations of 192 sign designs were summed to compute the average ratings for four design groups (with/without saliency feedback × urban/rural settings). We conducted a multi-way mixed analysis of variance (ANOVA) to examine the main and interaction effects of saliency, environment, and age. As all three factors had only two levels, the ANOVA also indicated the difference between the two levels (Figure 11). Descriptive statistics and full results of the ANOVA are presented in Appendices A and B.

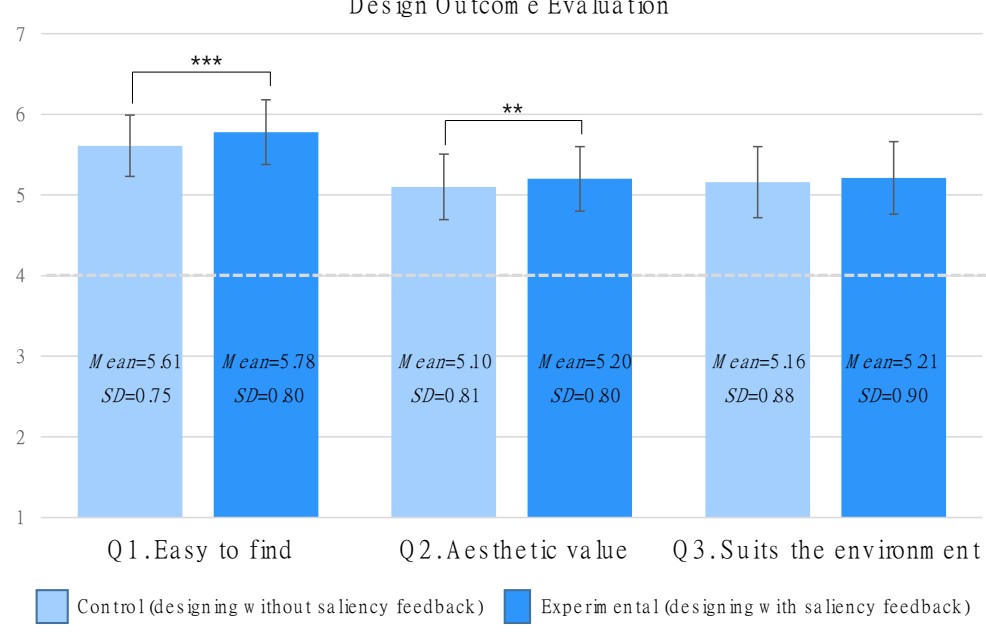

**Figure 11.** Perceived quality of signs produced with and without saliency feedback. Visual saliency helped to produce easier-to-find and more aesthetically pleasant designs (** $p < 0.01$, *** $p < 0.001$).

### 3.2.1. Designers Produced Better Designs When Provided with Saliency Predictions (RQ1)

Designs produced with saliency feedback were rated higher for the measures "easy to find and eye-catching" and "aesthetic value" than those produced without it. The participants evaluated the designs produced with visual saliency ($M = 5.78$, $SD = 0.080$) as easier to find than the designs from control groups ($M = 5.61$, $SD = 0.76$), with $F = 24.846$, $p < 0.001$, and $\eta^2_p = 0.213$. They also rated the saliency-assisted designs ($M = 5.20$, $SD = 0.80$) as more aesthetically pleasing than signs designed by the control group ($M = 5.10$, $SD = 0.81$), with $F = 8.800$, $p = 0.004$, and $\eta^2_p = 0.087$. The designs produced with the saliency map were also rated higher for Q3, i.e., "suits the environment", ($M = 5.21$, $SD = 0.90$) than those produced without it ($M = 5.16$, $SD = 0.88$), although the difference was not statistically significant ($F = 2.780$, $p = 0.099$, $\eta^2_p = 0.029$). In summary, the provision of saliency prediction in the urban design process enhanced the "easy to find" and "aesthetic value" aspects of the design quality, while maintaining environmental harmony, confirming RH1.

Conspicuous designs, as well as designs that focus excessively on aesthetics, can negatively impact the surrounding environment. The significant improvements in the first two measures are particularly meaningful because they were not accompanied by reductions in design reconcilability (Q3, "suits the environment").

The saliency feedback also enhanced the perceived aesthetic value (Q2). We believe that this result was partly due to the aesthetic–usability effect [77]; i.e., people find aesthetically pleasing things more usable. Additionally, under usability manipulation, the effect can be reversed; i.e., "people find usable things more beautiful" [78]. The difference in the usability of the signs, as measured through Q1, may have affected the perception of their aesthetics and caused the difference between conditions in Q2. The multiple linear regression analysis supported a significant effect between Q1 and Q3 on Q2, $R^2 = 0.684$, $F (2, 380)$ $= 408.699$, and $p < 0.001$. The individual predictors were examined further, indicating that Q1 ($t = 28.216$, $p < 0.001$) and Q3 ($t = 2.992$, $p = 0.003$) were predictors of Q2, implying that participants preferred the aesthetics of the saliency-guided designs due to their visibility.

### 3.2.2. The Enhancement in the Design Quality Due to Saliency Feedback Was More Significant in Urban Areas than in Rural Areas (RQ2)

The environment mainly affected Q1, i.e., "easy to find", suggesting that rural signs ($M = 5.80$, $SD = 0.84$) are more visible than urban signs ($M = 5.59$, $SD = 0.79$), with $F = 24.120$, $p < 0.001$, and $\eta^2_p = 0.208$ (Figure 12). This is consistent with the results of Costa et al. [44], who reported that drivers fixate more on road signs in the countryside than those in urban settings. We mainly attribute the result to visual distractions from other urban scenery or the lack of distractions in rural environments, which could have caused the evaluators to focus less on the designs in urban areas.

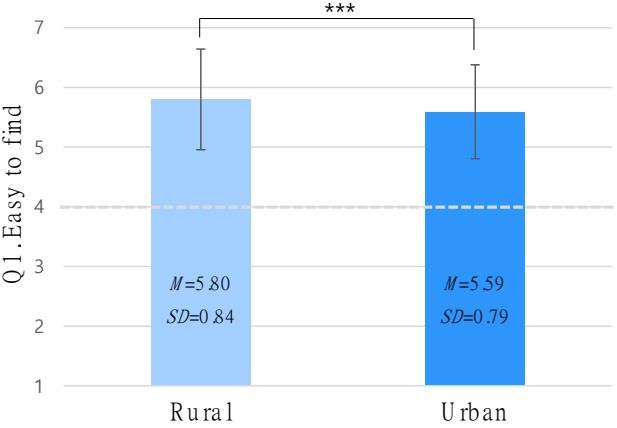

**Figure 12.** The environment's main effect on Q1. Signs in rural areas were easier to find (*** $p < 0.001$).

The interaction between the provision of saliency feedback and the environment also had significant effects for Q1 ("easy to find") and Q2 ("aesthetic value"), with $F = 19.055$, $p < 0.001$, and $\eta^2_p = 0.172$ and $F = 5.110$, $p = 0.026$, and $\eta^2_p = 0.208$, respectively. The slopes in Figure 13 indicate the effects of the interaction between the provision of saliency feedback and the environment on Q1 and Q2. For both Q1 ("easy to find", Figure 13a) and Q2 ("aesthetic value", Figure 13b), the disparity between the urban and rural environments decreased when the signs were designed with visual saliency. The saliency feedback improved the sign design quality to a greater degree in urban areas than in rural areas, which tended to contain fewer distractions, confirming RH2.

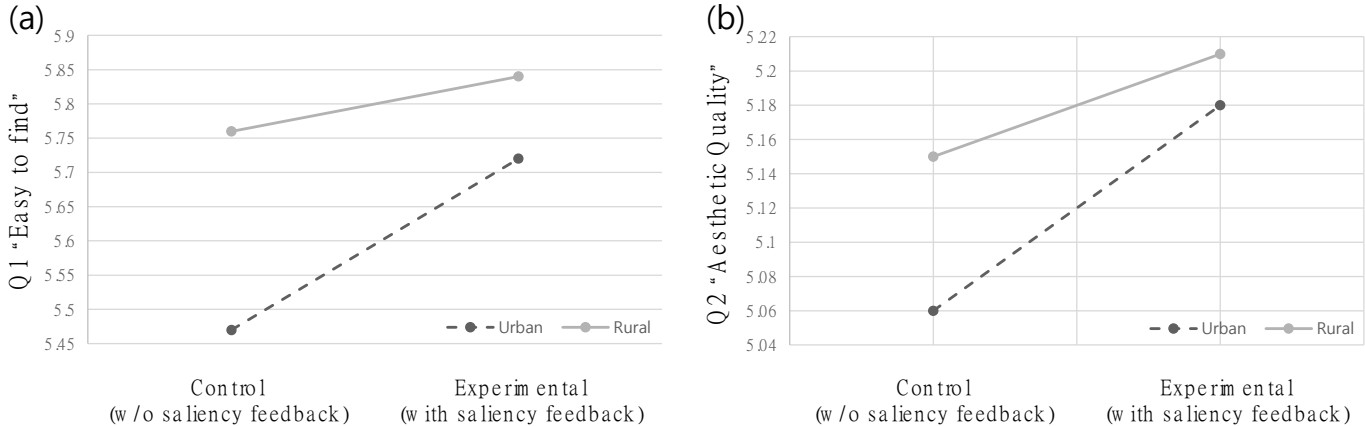

**Figure 13.** Interaction effect between the provision of saliency feedback during design and the environment on (**a**) Q1 ("easy to find") and (**b**) Q2 ("aesthetic quality"). The tool improved the conspicuity and aesthetics—particularly in urban areas. (Control: sign designs that were produced without prediction function, experimental: sign designs that were produced with saliency prediction function.)

This interaction effect was mainly due to visual distractions, which degraded the perceived design quality in two ways. First, distractions made it more difficult for the designers to predict where users would direct their attention. This was reported by control-group participant P2: "*I had little trouble in finding proper design and position of signs in rural areas, but in urban environments, other city elements like signs, buildings, cars, and streetlights made it harder to expect whether my design attracts pedestrians' eyes.*" Second, distractions make it more difficult for pedestrians to pay proper attention to signs when they are not carefully designed. Therefore, saliency feedback can result in better designs—particularly in urban settings with abundant distractions.

### 3.2.3. Saliency Feedback Significantly Enhanced the Design Quality Regardless of the User's Age or Cognitive Ability, but the Improvement in Aesthetics Was Particularly Appreciated by the Elderly with SCD (RQ3)

Age had no main or interaction effect on any of the three questionnaire items. However, an interaction effect between the provision of saliency feedback during the design process and SCD on Q2, i.e., "aesthetic value", was observed, with $F = 5.038$, $p = 0.027$, and $\eta^2_p = 0.052$ (Figure 14). Older adults with SCD preferred designs produced with visual-saliency prediction to the control designs more strongly than those without SCD. This preference is attributed to the aesthetic–usability effect [77]. The utility and conspicuity of signs designed with saliency prediction may have caused them to be perceived as more aesthetically pleasant. The results indicate that elderly people with SCD can benefit the most from designs produced with visual saliency, confirming RH3. Thus, saliency feedback can facilitate the design of more navigable cities.

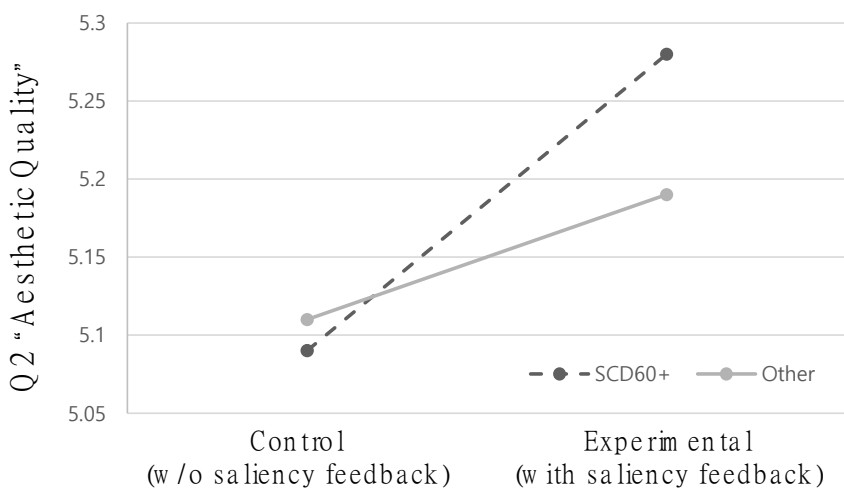

**Figure 14.** Interaction effect of the provision of saliency feedback during design and SCD on Q2. Older adults with SCD particularly appreciated the aesthetics of signs crafted with saliency, compared with other subjects' appraisals. (Control: sign designs that were produced without prediction function, experimental: sign designs that were produced with saliency prediction function.)

## 4. Discussion

### 4.1. Virtual- and Mixed-Reality Settings of Urban Design Simulation

Our proof-of-concept tool can be used in both virtual-reality settings with fully graphical backgrounds and camera-based mixed-reality environments at a target location. Luigi et al.'s [87] VR urban planning study found that people perceive visual features of a target area similarly in both actual and graphical settings, suggesting that experiments in both mixed and virtual reality can provide valid and transferable results. Table 3 summarizes the advantages and disadvantages, which should be carefully considered when integrating the tool into actual urban design simulation processes. For example, the graphical VR environment used in our user studies allows for natural pedestrian locomotion and greater flexibility in setting up the target environment. In contrast, the mixed-reality version of Urban Salviz creates photorealistic scenes without the need for extra work to create comparable 3D environments, making it easier for non-designers and small projects with limited funds for graphical environment development. However, the mixed-reality version does not provide a 3D representation of the target region, so designs cannot be simulated from as many pedestrian perspectives as in VR simulations. Instead, designers must rely on still photos or a few distinct images taken from different positions, which can still be effective enough when simulated at key intersection nodes of wayfinding decisions. Since our focus was on the situational and temporal context and visual attention that change rapidly as people travel, we used the virtual-reality version of Urban Salviz in the user study experiment.

**Table 3.** Advantages and disadvantages of our method and previous methods of urban design feedback.

| Settings | Components | Advantages | Disadvantages |
|---|---|---|---|
| Mixed reality | Computer USB camera for MR settings | Rapid construction of the ecologically valid environment Consistent to saliency dataset | Difficult to simulate the pedestrian's visual interaction with the surroundings |
| Virtual reality | Computer (+ digital twin 3D modeling) | Dynamic simulation from the virtual pedestrian's viewpoint (e.g., walk, stop, watch) | Difficult to include natural visual distractions or on-site events in the experiment |

*4.2. Providing Visual Saliency as Aid Enhanced UX of Designer and Design Quality*

While there was no difference in the overall usability between the conditions with and without saliency prediction, designers preferred designing with visual saliency, as it provided a more "helpful" and "efficient" design experience. Thus, utilities for attention prediction can be integrated with existing design tools without sacrificing usability.

In the user studies, the wayfinding signs designed using Urban Salviz with visual-saliency feedback received significantly higher scores for the "easy to find" measure than those designed without it. The effect magnitude indicated that the saliency prediction significantly improved the sign visibility ($\eta^2_p$ = 0.213). This is likely because designers were able to optimize their designs via attention prediction. While research has been performed on the use of saliency in graphic design tools, this study is the first to demonstrate that attention prediction can enhance the quality of urban designs, for which the spatial layout and the harmony between elements and the environment are important considerations.

The "aesthetic value" metric also exhibited significant differences, although we did not provide tools to evaluate the aesthetic value of signs. This outcome is mainly attributed to the aesthetic–usability effect [77], which may be inverted as "what is beautiful is what is useful" [78]. People evaluated the aesthetics of sign designs produced with saliency prediction more favorably because they were easier to discover, which is a crucial aspect of a sign's utility. However, the effect size was modest ($\eta^2_p$ = 0.087). For producing more attractive designs, a comprehensive tool should contain additional features such as color choices, layout modification, and the ability to evaluate a design's aesthetic value.

*4.3. Saliency-Guided Urban Design Was Particularly Effective in Urban Areas and for Elderly People with SCD*

The tool was more effective in urban areas than rural regions, especially among older people with SCD. When it is difficult to locate a sign due to severe visual distractions, the tool might be very useful. Additionally, the preference among the elderly with SCD is attributable to the fact that persons with cognitive decline often have diminished spatial navigation abilities. Indeed, 68% of surveyed participants with SCD complained about their spatial navigation skills [49]. Thus, our technique can make cities walkable for senior citizens by promoting more conspicuous urban elements design that helps the elder navigation.

*4.4. Limitations*

4.4.1. Selecting an Appropriate Colormap for Saliency Visualization Can Improve Usability

Colormaps have been widely studied for their role in data visualization [39,88] and are commonly used to visualize aggregated gaze data. Yet, rainbow colormaps are not recommended for data visualization due to their lack of perceptual uniformity and potential for misinterpretation [89]. Single-hue colormaps with a brightness scale are generally preferred because they offer clear perceptual ordering and are consistent across cultures [90]. Breslow et al. [91] compared the benefits and drawbacks of multi-hue and single-hue colormaps, finding that multi-hue colormaps are highly discernible and allow for fast serial searches of a legend, but are less perceptually ordered, while single-hue colormaps have clear perceptual ordering but are less discernible. Liu et al. [39] also noted that single-hue colormaps may provide insufficient resolution at a smaller span, making multi-hue colormaps preferable for visualizing scalar fields such as heatmaps. There is no consensus on the best colormap for visualizing visual saliency, but a proper choice can improve usability and enhance the design experience with clear information on pedestrian attention.

4.4.2. Placebo Effect Was Not Controlled in Our Experiment Design

To more accurately compare the impact of visual-saliency prediction in the experimental condition to the control condition without this function, it is important to control for placebo feedback. While providing saliency prediction as feedback has been shown to

improve the design experience and quality of designs by novice designers, it is possible that the feedback itself, regardless of its accuracy, may have influenced their experience and resulting quality. The placebo effect, which is well-known in biomedical research, has not been widely discussed in design research. However, a user experience study by Vaccaro et al. [92] found that the placebo effect can also occur in UX experiments, with people feeling more satisfied with control settings even when they do not function properly.

*4.5. Future Work*

4.5.1. Implementation of Urban Salviz with More Universal Dataset

Urban Salviz was fed realistic renderings of urban scenery in a game engine to generate a saliency map. However, the model was trained on real images from the SALICON dataset for compatibility with both virtual and mixed-reality environments. Despite this limitation, the VR user study showed that Urban Salviz effectively enhances design experience and quality. To further improve the system's capability and effectiveness, the saliency prediction model could be trained on separate datasets containing either real or graphical images. Sitzmann et al. [93] have established the foundations for saliency in virtual-reality environments and enhanced saliency prediction accuracy with a saliency dataset on VR scenes may be beneficial. However, the transition from real to graphical images should be carefully considered to preserve the ecological validity of the system.

Our results suggest that saliency-guided design is particularly effective for the elderly with SCD. However, the SALICON dataset used in our study was collected from subjects aged 19–28 years and may not accurately represent the entire population. If the model is trained using datasets from specific groups (e.g., aging adults, children, or people with low vision), the tool can be even more useful for designing inclusive cities.

4.5.2. Tools and User Studies for Other Specific Urban Design Tasks and Expertise Levels

Visual-saliency feedback can be used in various urban design domains where the impact of visual elements must be evaluated in advance along with their surroundings (e.g., architecture, landmarks, posters). Nonetheless, this user study is limited to wayfinding signs and this could be expanded. Future user tests involving additional design tasks will reveal how Urban Salviz must be modified to fit the relevant domain. For instance, the tool is suited for the pre-design or design development phases of architecture design, where simulations of basic concept development and design details are required. Most architecture design is considerably more time-consuming to complete, hence the frequency with which designers examine or rely on the saliency prediction may differ. Additionally, the proposed tool can be used to design urban components that should be displayed minimally (e.g., architectures designed not to deteriorate landscape, power generators, and water pipes). Indeed, how negligible a design is an important criterion to evaluate wind turbine generators [25] or solar panels in building retrofit projects [2]. In such cases, an alternative method may be required to illustrate how "not noticeable" a design draft is.

Despite the increased requirement for participatory planning, urban design is still predominantly decided by professional designers. While novice designers lack confidence in their designs, expert designers rely on their experience and intuition to evaluate designs [57]. They employ a preliminary evaluation procedure prior to and throughout the design implementation [57], which may be consistent or inconsistent with saliency prediction. Expert usage of the proposed tool will show how the tool can be integrated with existing processes. Further user studies may reveal whether experts agree with and adopt the feedback or prefer traditional methods of evaluation. Examples of research topics that can be explored include the number of iterations and the time needed for each design decision when using saliency feedback, the level of trust and reliance on the feedback by designers, and design preferences when working with saliency feedback. This research can provide a deeper understanding of the interaction between designers and saliency feedback and inform how the saliency feedback should be integrated with existing tools.

## 5. Conclusions

Urban form and architectural components should be designed to attract or divert attention as they contribute to the appearance and functionality of a city's environment. In particular, wayfinding signs should be sufficiently noticeable without harming the overall landscape, as accessible wayfinding is essential for inclusive navigation, and these signs play a key role in route guidance. However, similar to other urban elements, sign visibility is a complex construct that involves sign design, context, and interaction with moving pedestrians. Attention to wayfinding signs varies according to location or as the visual scene changes during navigation. Thus, wayfinding signs should be iteratively simulated in the target environment to improve the design with regard to pedestrian attention. Currently, there is no method to comprehensively analyze users' attention in cities; thus, urban designers rely on heuristics for either salient or inconspicuous design, limiting their options. To address this gap, we incorporated visual saliency that predicts pedestrian attention to simulate the visual interactions of pedestrians with wayfinding signs, landmarks, and other city elements. Designers can iteratively simulate the shape, size, color, and text of their sign design while observing how the modifications affect users' attention.

Our study provides practical insight into saliency-assisted urban design methods. First, to incorporate the saliency prediction into urban design, we conducted a case study on wayfinding design to evaluate the benefits of this design method regarding three aspects: (1) the effects of saliency feedback on the perceived usability and design experience, (2) its effect on the quality of sign design, and (3) how end users of different demographics perceived signs designed using saliency feedback. Our findings indicated that providing visual saliency as iterative feedback during the urban design process can enhance the design experience without negatively impacting usability, making it a viable option for urban design. We also found that signs produced with saliency feedback were perceived as easier to find and more aesthetically pleasing, indicating that this method promotes more noticeable and aesthetically pleasing outcomes without compromising the design's environmental harmony. Thus, visual-saliency feedback can be used to help designers predict the attention of pedestrians in cities and simulate dynamic interactions between moving pedestrians and urban elements. The results of this study can be used to facilitate navigation with conspicuous and aesthetically pleasant signs and landmarks, which can make cities more accessible and inclusive—particularly for people who find wayfinding difficult, such as those unfamiliar with an area or older adults with cognitive decline.

**Author Contributions:** Conceptualization, G.K.; methodology, G.K. and S.K.; software, G.K. and D.Y.; formal analysis, G.K.; data curation, G.K. and J.L.; writing—original draft preparation, G.K.; writing—review and editing, G.K., D.Y., J.L. and S.K.; supervision, S.K. All authors have read and agreed to the published version of the manuscript.

**Funding:** This research was supported in part by the National Research Foundation of Korea (NRF) funded by the MSIT (2021R1A4A1030075), in part by the the Korea Institute of Energy Technology Evaluation and Planning (KETEP), the Ministry of Trade, Industry and Energy (MOTIE) of the Republic of Korea (No. 20204010600340), and in part by the GIST-MIT Research Collaboration grant funded by the GIST in 2023.

**Institutional Review Board Statement:** The study was conducted according to the guidelines of the Declaration of Helsinki and approved by the Institutional Review Board of Gwangju Institute of Science and Technology (Protocol code 20190510-HR-45-02-02, approved on 10 May 2019).

**Informed Consent Statement:** Informed consent was obtained from all subjects in the study.

**Data Availability Statement:** The data presented in this study are available upon request.

**Conflicts of Interest:** The authors declare no conflict of interest.

## Appendix A. Descriptive Statistics for Design Quality Assessment

**Table A1.** Descriptive statistics of the crowd ratings for design quality assessment item Q1 ("easy to find").

| Q1 | | Experimental (Designing with Saliency Feedback) | | | | |
| --- | --- | --- | --- | --- | --- | --- |
| | | Age | | SCD | | Total |
| | | Older | Younger | SCD60+ | Others | |
| | Urban | 5.83 | 5.68 | 5.86 | 5.7 | 5.72 |
| Environment | Rural | 5.99 | 5.79 | 6.12 | 5.8 | 5.84 |
| | Total | 5.92 | 5.73 | 5.99 | 5.75 | 5.78 |
| | | Control (Designing without Saliency Feedback) | | | | |
| | | Age | | SCD | | Total |
| | | Older | Younger | SCD60+ | Others | |
| | Urban | 5.59 | 5.43 | 5.5 | 5.46 | 5.47 |
| Environment | Rural | 5.98 | 5.68 | 6.09 | 5.88 | 5.76 |
| | Total | 5.78 | 5.56 | 5.8 | 5.67 | 5.61 |

**Table A2.** Descriptive statistics of the crowd ratings for design quality assessment item Q2 ("aesthetic value").

| Q2 | | Experimental (Designing with Saliency Feedback) | | | | |
| --- | --- | --- | --- | --- | --- | --- |
| | | Age | | SCD | | Total |
| | | Older | Younger | SCD60+ | Others | |
| | Urban | 5.3 | 5.14 | 5.26 | 5.17 | 5.18 |
| Environment | Rural | 5.28 | 5.19 | 5.3 | 5.2 | 5.21 |
| | Total | 5.29 | 5.17 | 5.28 | 5.19 | 5.2 |
| | | Control (Designing without Saliency Feedback) | | | | |
| | | Age | | SCD | | Total |
| | | Older | Younger | SCD60+ | Others | |
| | Urban | 5.16 | 5.03 | 4.92 | 5.08 | 5.06 |
| Environment | Rural | 5.29 | 5.16 | 5.26 | 5.13 | 5.15 |
| | Total | 5.23 | 5.09 | 5.09 | 5.11 | 5.1 |

**Table A3.** Descriptive statistics of the crowd ratings for design quality assessment item Q3 ("suits the environment").

| Q3 | | Experimental (Designing with Saliency Feedback) | | | | |
| --- | --- | --- | --- | --- | --- | --- |
| | | Age | | SCD | | Total |
| | | Older | Younger | SCD60+ | Others | |
| | Urban | 5.41 | 5.14 | 5.18 | 5.21 | 5.21 |
| Environment | Rural | 5.43 | 5.2 | 5.25 | 5.26 | 5.26 |
| | Total | 5.42 | 5.17 | 5.21 | 5.24 | 5.23 |
| | | Control (Designing without Saliency Feedback) | | | | |
| | | Age | | SCD | | Total |
| | | Older | Younger | SCD60+ | Others | |
| | Urban | 5.33 | 5.11 | 5 | 5.19 | 5.16 |
| Environment | Rural | 5.43 | 5.15 | 5.25 | 5.21 | 5.21 |
| | Total | 5.38 | 5.13 | 5.13 | 5.2 | 5.19 |

## Appendix B. Multi-Way Mixed ANOVA for Design Quality Assessment

**Table A4.** Main and interaction effects of the provision of saliency feedback during the design process, the environment, the participant age, and SCD on end-user ratings (design quality evaluation) for the sign design outcome that was produced in the user study (multi-way mixed ANOVA).

| Source | df | F | $p$ | Effect Size ($\eta^2_p$) |
|---|---|---|---|---|
| **Q1. Easy to find** | | | | |
| Saliency feedback | 1 | 24.846 *** | <0.001 | 0.213 |
| Saliency feedback × Age | 1 | 1.868 | 0.175 | 0.020 |
| Saliency feedback × SCD | 1 | 1.410 | 0.238 | 0.015 |
| Environment | 1 | 24.120 *** | <0.001 | 0.208 |
| Environment × Age | 1 | 0.016 | 0.898 | 0.000 |
| Environment × SCD | 1 | 2.417 | 0.123 | 0.026 |
| Saliency feedback × Environment | 1 | 19.055 *** | <0.001 | 0.172 |
| Saliency feedback × Environment × Age | 1 | 0.087 | 0.768 | 0.001 |
| Saliency feedback × Environment × SCD | 1 | 2.282 | 0.134 | 0.024 |
| Error | 92 | | | |
| **Q2. Aesthetic value** | | | | |
| Saliency feedback | 1 | 8.800 ** | 0.004 | 0.087 |
| Saliency feedback × Age | 1 | 3.654 | 0.059 | 0.038 |
| Saliency feedback × SCD | 1 | 5.038 * | 0.027 | 0.052 |
| Environment | 1 | 2.546 | 0.114 | 0.027 |
| Environment × Age | 1 | 1.422 | 0.236 | 0.015 |
| Environment × SCD | 1 | 2.936 | 0.090 | 0.031 |
| Saliency feedback × Environment | 1 | 5.110 * | 0.026 | 0.053 |
| Saliency feedback × Environment × Age | 1 | 0.007 | 0.936 | 0.000 |
| Saliency feedback × Environment × SCD | 1 | 3.011 | 0.086 | 0.032 |
| Error | 92 | | | |
| **Q3. Suits the environment** | | | | |
| Saliency feedback | 1 | 2.780 | 0.099 | 0.029 |
| Saliency feedback × Age | 1 | 0.211 | 0.647 | 0.002 |
| Saliency feedback × SCD | 1 | 0.611 | 0.436 | 0.007 |
| Environment | 1 | 1.148 | 0.287 | 0.012 |
| Environment × Age | 1 | 0.252 | 0.617 | 0.003 |
| Environment × SCD | 1 | 0.820 | 0.367 | 0.009 |
| Saliency feedback × Environment | 1 | 1.130 | 0.291 | 0.012 |
| Saliency feedback × Environment × Age | 1 | 0.029 | 0.865 | 0.000 |
| Saliency feedback × Environment × SCD | 1 | 1.223 | 0.272 | 0.013 |
| Error | 31 | | | |

\* $p < 0.05$, \*\* $p < 0.01$, \*\*\* $p < 0.001$.

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
