# Peer review of "Simulating Urban Element Design with Pedestrian Attention: Visual Saliency as Aid for More Visible Wayfinding Design"

_land, doi:10.3390/land12020394_

Round 1

Reviewer 1 Report

This is an original work that deals with an issue that is generally little addressed in scientific research: the design of urban elements. A conceptual and methodological effort is made to try to present an original computer tool to support the design of urban elements (Urban Salviz). 

Despite this, the text is long, complex and difficult to understand and follow. The work provides an excess of unstructured information. The objectives of the work, its methodology and the results obtained are not clear. The analysis of the results is formalised in the analysis of the usability of the application and in the analysis of the quality of the design based on interaction with users.

It should be understood that this is the design of a computer application that is developed on a Python programming environment and that makes use of the tools of the Unity 3D gaming platform.

The work requires major changes to be published. The objectives of the work need to be clear from the outset. It should be made clear whether a methodology is being proposed or whether the capabilities of a software application are being evaluated.

From the scientific point of view, it is important to clearly explain the methodology developed for the resolution of each of the scientific problems posed by the design of urban elements. It is not enough to present a screenshot with the functionalities of an application.

The article must be restructured in a comprehensive way following the structure: Introduction/Methods/Results/Discussion/Conclusions. 

A methodological outline must be included to clarify what the methodology of the study consists of.  Each of the methodological solutions solved by the application must be clear what they consist of.  The technological architecture should also be explained in greater detail and include clear development schemes.

The figures should be rigorously explained, as there are pictures and images that are not easy to interpret. In many cases they appear redundant.  Each of the elements of the work must be well described.

It would be advisable to include a table of contributions and references for the analysis of the state of the art to show what kind of solutions currently exist to solve urban design. In addition, it must be clear what the real contribution of the project is. It must be explained in detail what Salicon contributes to the project.

From the point of view of reproducibility of the work, it should be explained how the programme has been compiled and whether it can be reused by the scientific community.

The two application testing analyses carried out should be summarised in a table in order to clearly appreciate the advantages of their implementation.

Author Response

We are grateful to reviewer 1 for the valuable feedback. We have made revisions to the manuscript in response to the comments and have attached a document highlighting these changes for your reference.

Reviewer 2 Report

The scientific contributions and innovative points should be well summarized and improved. A meaningful research question should be further clarified. It can be seen from the paper that the main contents include two parts: presenting a design tool (Urban Salviz) and then proving its usefulness. The developing process, core models, and related data are supposed to be the key points if the authors would like to fill the gap that there is no such tool available, but the authors just “present” functionalities and implementation details of Urban Salviz. As a result, it seems that the development of the tool was not important and the authors mainly want to assess its usefulness in this paper. However, assessing the usefulness of the authors’ tool is not a significant research question, because the authors have not established a solid basis that the tool is scientifically developed and that the usefulness of this tool is a common concern of the academic community.

I suggest that the authors re-consider the points of this paper. If the authors want to verify the usefulness of your tool, it is crucial to firstly elaborate how the tool is built to solve the existing issues with wide interest and illustrate if the tool is developed rigorously. If the authors want to reveal how design quality is impacted when designing with visual saliency, this paper should be re-organized by highlighting the “user study design” as an independent Methods section and the use of Urban Salviz should be only regarded as a case study. The introduction on the tool itself is no longer important, while the rationale for selecting Urban Salviz really matters. At the end, the conclusion should be therefore more generalized beyond any specific tools.

Author Response

We are grateful to reviewer 2 for the valuable feedback. We have made revisions to the manuscript in response to the comments and have attached a document highlighting these changes for your reference.

Reviewer 3 Report

Recommendations:

- The paper introduces an interesting urban design support tool -wayfinding  "Urban Salviz", a mix of both mixed-reality and virtual reality options. Therefore, the clarity of the images can be enhanced.

- 98 references are listed which I think too many. Therefore, I suggest to reduce.

- The authors' acknowledgement of the paper limitation is notable and reflects the need for further studies on the same topic.

Author Response

We are grateful to reviewer 3 for the valuable feedback. We have made revisions to the manuscript in response to the comments and have attached a document highlighting these changes for your reference.

Round 2

Reviewer 1 Report

The paper has been corrected and improved.

Author Response

Dear reviewer 1,

We are submitting a revision of our earlier manuscript, “Simulating Urban Elements Design with Pedestrian Attention: Visual Saliency as an Aid for More Visible Wayfinding Design”

We are thankful to Reviewer 1 for the constructive feedback. We are especially appreciative of their thorough review of our response and the revised paper. We belive our manuscript has been enhanced as a result of addressing Reviewer 1's comments.

Thank you for your consideration.

Sincerely,

Gwangbin Kim, Dohyeon Yeo, and Jieun Lee

Ph.D. Program, School of Integrated Technology, Gwangju Institute of Science and Technology

SeungJun Kim, Ph.D.

Associate Professor, School of Integrated Technology, Gwangju Institute of Science and Technology

Reviewer 2 Report

The manuscript is still too long.

The first RH is that providing visual-saliency prediction during the design of urban elements would enhance the quality of the designs. Regarding this, I have several questions. 1) why is it necessary to address this hypothesis? Does it remain unclear in the literature? In my opinion, it is quite obvious that visual-saliency prediction can improve designs. Comparing to knowing whether a method is good or not, readers may want to learn more about how it works, why it is good, and if it can be improved further. 2) How do the authors define and measure “the enhanced quality of the designs”? It is not clear how the study design in section 2.3 and 2.4 supports the test of the three hypotheses. Line 330-332: Can the authors present one or two questions in the questionnaire? 3) Can Urban Salviz be generalized as visual-saliency prediction? What is the relationship between saliency-guided iterative design and Urban Salviz?

I suggest that the authors summarize a framework of the saliency-guided iterative design which can be applied in all practical tools including Urban Salviz. Figure 1 is ok but it can be improved by illustrating more underlying models and workflows. Saliency map, machine learning models, and other models of modules can be integrated into the figure. The authors may not introduce these models in the section of Urban Salviz. Urban Salviz is not important and it is only a software to conduct the saliency-guided iterative design. The authors can still do the user study and design quality evaluation by comparing the two design methods; one has saliency prediction module while the other one does not. You can state that both the two comparable methods are conducted by using Urban Salviz.

Author Response

We appreciate reviewer 2 for the constructive feedback. Please find the attached file for our response to reviewer 2.
